# A spatial regime shift from predator to prey dominance in a large coastal ecosystem

Johan S. Eklöf [1✉], Göran Sundblad [2], Mårten Erlandsson[3], Serena Donadi [1,2], Joakim P. Hansen [4], Britas Klemens Eriksson[5,6] & Ulf Bergström[3,6]

Regime shifts in ecosystem structure and processes are typically studied from a temporal perspective. Yet, theory predicts that in large ecosystems with environmental gradients, shifts should start locally and gradually spread through space. Here we empirically document a spatially propagating shift in the trophic structure of a large aquatic ecosystem, from dominance of large predatory fish (perch, pike) to the small prey fish, the three-spined stickleback. Fish surveys in 486 shallow bays along the 1200 km western Baltic Sea coast during 1979–2017 show that the shift started in wave-exposed archipelago areas near the open sea, but gradually spread towards the wave-sheltered mainland coast. Ecosystem surveys in 32 bays in 2014 show that stickleback predation on juvenile predators (predator–prey reversal) generates a feedback mechanism that appears to reinforce the shift. In summary, managers must account for spatial heterogeneity and dispersal to better predict, detect and confront regime shifts within large ecosystems.

[1] Department of Ecology, Environment and Plant Sciences, Stockholm University, Stockholm, Sweden. [2] Department of Aquatic Resources, Swedish University of Agricultural Sciences, Drottningholm, Sweden. [3] Department of Aquatic Resources, Swedish University of Agricultural Sciences, Öregrund, Sweden. [4] Stockholm University Baltic Sea Center, Stockholm, Sweden. [5] Groningen Institute for Evolutionary Life-Sciences, University of Groningen, Groningen, The Netherlands. [6]These authors contributed equally: Britas Klemens Eriksson, Ulf Bergström. ✉email: johan.eklof@su.se

Over the last half century, ocean and coastal ecosystems have been increasingly observed to sometimes shift unexpectedly to alternative and seemingly persistent sets of dominating species and processes (regimes) in response to environmental and biotic changes[1,2]. In parallel, theoretically driven mathematical models triggered the intriguing idea that ecosystems can display multiple stable states under similar conditions[3,4]. Following intense debates regarding the evidence for such "multi-stability" in the real world[5], the reconciliation of observation and theory under the umbrella concept "regime shifts"—which we here define as abrupt and long-term changes in ecosystem structure and functions, including shifts between stable states[1,6–8]—facilitated an exponential growth[9] in efforts to identify, predict and reverse ecosystem shifts across the world's biomes[1,2,6,9,10]. Today we know that regime shifts are typically caused by external "shocks" or gradually changing environmental conditions that exceed critical thresholds (a.k.a. "phase shifts"), but that some also involve *critical transitions*[1,2,10] where novel feedbacks propel the system from one self-reinforcing and persistent regime (or stable state) to another[3,10]. All regime shifts are difficult to manage, but critical transitions pose particular challenges because of inherent difficulties in both predicting and reversing them[10].

Despite well-developed theory and many empirical examples, we are still far from understanding and managing regime shifts. One particular challenge is that most studies ignore the role of spatial variability[11–14]; a paucity stemming from the fact that many model systems (e.g. shallow lakes) are relatively small, homogenous and have hard physical boundaries. However, recent advances suggest that large, heterogeneous ecosystems with permeable boundaries (e.g. grasslands or coastal sea areas) can instead display *spatial regimes*, i.e. spatially explicit sets of similar structures and functions maintained by self-reinforcing feedback mechanisms within their boundaries[15,16]. Theory predicts that in such systems, gradual environmental change can trigger the initiation of local (patch-level) sudden shifts in ecosystem structure, that are asynchronous because of the spatial heterogeneity[12]. Consequently, the whole-system response becomes gradual (the mean of many asynchronous small-scale shifts) rather than "catastrophic" (threshold-like)[12,13,17]. If organisms also disperse between the patches, the shift should start in areas where systems are closest to environmental thresholds, but then gradually spread as a traveling wave, front or falling dominos[12,13,17]—similar to spread of epidemic disease, financial crises and revolts in societies[10]. However, evidence for such spatial or "gradual"[13] regime shifts[4] in nature is limited to small-scale dynamics of desert vegetation[18] and decadal changes in the distribution of grassland bird assemblages[19].

In oceans and lakes across the world, declines of large predatory fish disrupt ecosystem functioning, economies, and human livelihoods[20–22]. By releasing smaller consumers from top-down control[23], predator decline can cause long-lasting shifts to prey-dominated regimes with profound impacts on ecosystem structure and function[1,24,25]. Such regime shifts are often caused by predator overharvest and/or gradual changes in environmental conditions[20–22], but can also be accelerated by novel feedback mechanisms, that may stabilize the alternative regime and prevent natural recovery[1,25,26]. One such feedback is predator–prey role reversal[27], where predator decline benefits small prey organisms that, in turn, prevent predator recovery by feeding on their early life stages[26,28]. Such role reversal has mainly been studied from a temporal perspective, but can in theory trigger traveling waves of prey dominance[29].

Simple predator–prey interactions can generate complex system-level phenomena (e.g. limit cycles[30]) that typically play out over time, but sometimes also over space. One of the most conspicuous examples is consumer fronts; hyperdense aggregations of mobile consumers (ranging from small zooplankton to large tropical ungulates) along resource edges[31]. The universal mechanism is that increasing consumer abundance leads to local overconsumption of resources, which triggers resource-dependent movement and, ultimately, the formation of fronts that propagate like traveling waves[32]. The fronts are typically transient or sometimes cyclic[32] and dissipate because food resources become limiting, and/or because consumers are dominated by large cohorts that naturally die or disperse[31]. However, theory[30] and observations[33] suggest that if the consumers are instead generalists that can switch to a more common prey, the fronts may lead to persistent regime shifts.

Here we describe a spatial shift from a predator- to prey-dominated marine ecosystem regime that has gradually spread through the 1200 km western Baltic Sea coast (northern Europe) over the last four decades. Starting in the early 1990s, abundances of a common mesopredatory fish—three-spined stickleback (*Gasterosteus aculeatus*, hereafter "stickleback")—rapidly increased nearly 50-fold in offshore areas[34]. Various evidence suggests that reduced predation pressure from declining stocks of large predatory fish increased stickleback survival[34–37], while eutrophication[38] and rapid ocean warming[39] increased their population growth[35,40]. Because adult stickleback migrate from the open sea to the coast in spring to breed, they effectively link coastal and offshore processes[35,37]. Along the coast, two large predatory fish species—Eurasian perch (*Perca fluviatilis*, "perch") and Northern pike (*Esox lucius*, "pike")—can locally control stickleback abundances[36]. However, predator abundances have declined along parts of the coast, potentially due to fisheries[41,42], increased seal and cormorant predation[42], and degradation of recruitment habitats[35,43]. Consequently, stickleback today dominate large coastal areas and there generate a trophic cascade that increases algal blooms, degrades habitat-forming benthic vegetation and exacerbates effects of eutrophication[35,36,44]. Recently, stickleback have also been shown to feed on perch and pike larvae, potentially suppressing their recruitment[34,43,45]. This indicates that predator–prey reversal may have reinforced a shift to a stickleback-dominated regime[34,35,37].

We hypothesized that strong environmental gradients across this coastal ecosystem facilitated a spatial regime shift from large (perch, pike) to small (stickleback) predatory fish dominance, gradually propagating from areas near the open sea towards the mainland coast (Fig. 1a). The reason is that stickleback historically reproduced mainly in wave-exposed bays in the outer parts of the vast (up to 70 km wide) and environmentally heterogeneous archipelago, whereas perch and pike reproduce mainly in wave-sheltered bays in the inner and middle archipelago[37,46]. Combined with evidence that both perch, pike and stickleback predation can structure local recruitment[43,45], this suggests that the middle archipelago should be bimodal with stickleback- or perch and pike-dominated bays. But as stickleback numbers gradually increased[34] and pike and perch have locally declined[2,28], stickleback dominance should gradually expand towards the coast.

Using a unique dataset of fish surveys in nearly 500 shallow bays over 39 years we confirm the existence of a spatial shift to stickleback dominance, gradually spreading from outer archipelago areas toward the mainland coast. The longest time series from a single area (Forsmark) shows that local shifts may be abrupt with clear temporal breakpoints, even though the large-scale spatial shift is gradual (the mean of many asynchronous local shifts). Finally, a detailed ecosystem survey in 32 bays shows that stickleback suppression of juvenile perch and pike (predator–prey role reversal) forms a strong feedback mechanism that appears to reinforce the shift. These findings

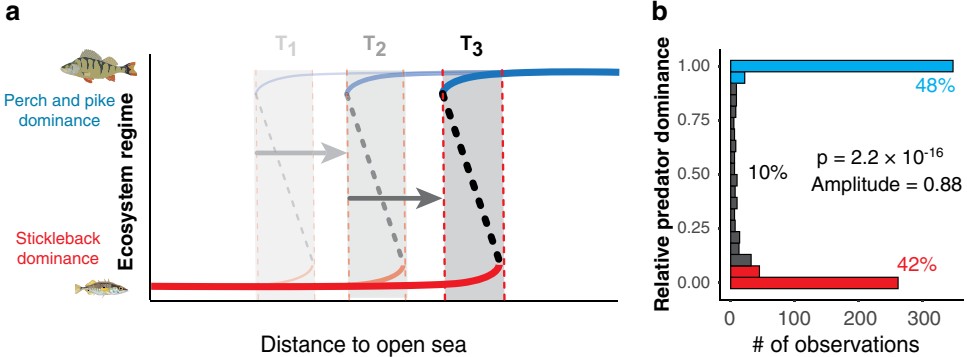

**Fig. 1 Spatial regime shift and bimodality in the coastal Baltic Sea ecosystems. a** Hypothesized spatial transition from a stickleback- to perch- and pike-dominated ecosystem regime with increasing distance to open sea at three points in time ($T_{1-3}$). Dotted vertical red lines are spatial breakpoints and shaded areas are bistable zones. **b** Histogram of relative predator dominance (pooling across all bay-years, $N = 829$); an index where $1 = 100\%$ dominance of juvenile perch and pike, and $0 = 100\%$ dominance of juvenile stickleback. P is the likelihood of unimodality based on Hartigan's dip test, and amplitude (range 0–1) the distinctness of the modes.

emphasize that detecting and confronting regime shifts in spatially extended ecosystems will require addressing underlying drivers while accounting for spatial heterogeneity and organism dispersal.

## Results

**Pooled data from multiple surveys reveal a regime shift over time and space.** To test our hypothesis we gathered fish abundance data from various research projects and monitoring programs that together covered sufficient spatial and temporal scales. The final dataset included 13,073 samplings of juvenile fish (young-of-the-year) in 477 shallow bays during 39 years (1979–2017), spread across 1200 km of the western Baltic Sea archipelago coast (55–65°N). Most (75%) of the bays were sampled once but some up to 34 years, resulting in 846 unique bay-year combinations. For each bay-year we then calculated the relative predator dominance (perch + pike abundance/perch + pike + stickleback abundance); a ratio from 1 to 0 where $1 = 100\%$ perch and pike dominance and $0 = 100\%$ stickleback dominance. As expected from theory (Fig. 1a), the ratio was highly bimodal with 48% predator (perch + pike) domination and 42% stickleback domination (here defined as bay-years with relative predator dominance $\geq 0.9$ and $\leq 0.1$, respectively; Fig. 1b). We then used binomial generalized linear regression to assess if and how relative predator dominance changed over time (year), space (distance to the open sea) and their interaction, while controlling for two covariates also affecting recruitment; wave exposure and latitude[37,46]. Since 90% of the observations belong to one of the two modes (Fig. 1b), this model also predicts the likelihood (0–1) that individual bays, as well as the proportion of sampled bays, were predator- or stickleback-dominated.

The best-fitting model confirms a gradual shift to stickleback dominance that propagated from the outer to the inner archipelago, evident by a strong interaction between time and distance to open sea, as well as main effects of distance, wave exposure and latitude (Fig. 2a–f, Supplementary Fig. 1, Supplementary Table 1). We find a positive influence of distance to open sea on relative predator dominance, with stickleback dominance and/or bimodality in outer archipelago areas, and predator dominance closer to the mainland coast (Fig. 2a–c). However, the slope of the curve decreased over time as bays increasingly shifted to stickleback dominance, starting near the open sea. Consequently, the distance from the open sea to the point at which local stickleback- or predator dominance were equally likely (red vertical lines in Fig. 2a–c) increased from ~8 km in 1996 to ~21 km in 2014. Likewise, the distance to the innermost

stickleback-dominated bay increased from ca. 8 to 26 km (Supplementary Fig. 2). As a consequence of these spatial dynamics, the timing of local shifts to stickleback dominance depended on the distance to the open sea (Fig. 2d–f): bays in the outer archipelago were initially bimodal but shifted to nearly complete stickleback dominance in the 1990s, whereas bays in the middle archipelago were predator-dominated until the early 2000s, after which most shifted to stickleback dominance. Finally, bays in the inner archipelago only started shifting in the early 2000s, but appear to be following the same trend as outer areas.

Following more typical regime shift studies[1] we also tested for a temporal shift from perch to stickleback dominance at the local scale, as evident by change-point(s)[47] in the longest time-series from an individual bay in our dataset; a 34-year sampling program outside Forsmark (60.4°N, 18.2°E; sampled 1981–2017). There was a change-point in year 2004, separating long-term perch dominance from stickleback dominance (Fig. 3a, Supplementary Fig. 3). Generalized additive models of perch and stickleback abundances (Fig. 3b) show that the shift was preceded by a rapid perch decline in the early 1990s ($F = 4.98$, $p = 0.001$) and an exponential stickleback increase starting in the early 2000s ($F = 5.85$, $p = 0.001$). This sequence of events supports the hypotheses that mesopredator release contributes to the rise of the stickleback[36]. Moreover, the 2004 breakpoint separating perch and stickleback dominance supports that even though the regional change in dominance is gradual (Fig. 2), local shifts can be abrupt.

Relative predator dominance is a ratio that in theory could respond to changes in only stickleback- or perch and pike numbers. Therefore, we also tested how the respective abundances of perch, pike and stickleback changed over time and space, using linear models (see Methods for details). Pooled perch and pike juvenile abundance declined exponentially over time across the entire coast, after accounting for a positive influence of distance to the open sea and a negative influence of wave exposure (Supplementary Fig. 4a–c, Supplementary Table 1). In contrast, stickleback juvenile abundance increased the most in wave-exposed bays in the inner archipelago along the southern coast, as supported by the significant interactions of time × distance, time × wave exposure and time × latitude (Supplementary Fig. 4d–f, Supplementary Table 1). The highest and temporally most stable stickleback abundances occurred in the outer archipelagos (Supplementary Fig. 4d–f).

**Predation and predator–prey reversal dictates local dominance.** To assess whether predator–prey reversal is strong enough to

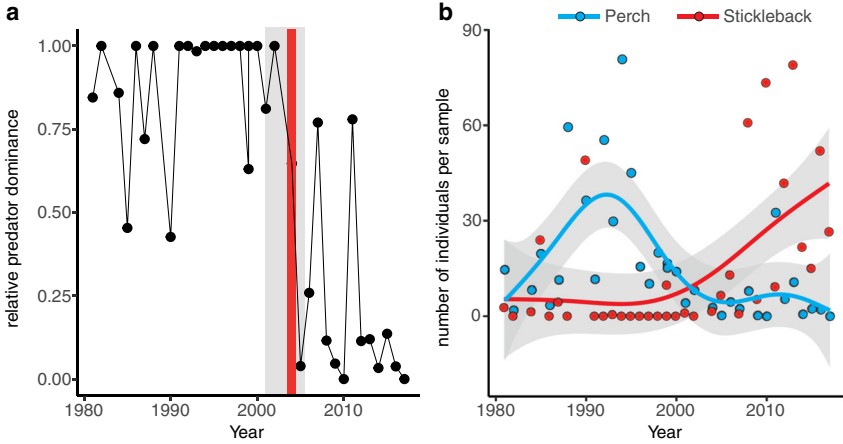

**Fig. 2 A gradual regime shift in space and time.** Partial regression plots showing relative predator dominance as a function of (**a**–**c**) distance to open sea at three snapshots in time (years 1996, 2008, 2014), and (**d**–**f**) time at three archipelago zones (outer, middle and inner), after controlling for the effects of wave exposure and latitude. Note that the points are partial residuals and therefore appear closer to the regression line than in reality. Vertical red lines show the spatial and temporal breakpoints at which relative predator dominance = 0.5, i.e. where local stickleback- and predator dominance are equally likely.

**Fig. 3 Local regime shift from perch to stickleback dominance.** Temporal changes in (**a**) relative predator dominance and (**b**) perch (blue) and stickleback (red) abundance, during 1981–2017 (no data from 1983, 1989, 2003) at Forsmark (60.4°N, 18.2°E). Red line in a) marks a temporal breakpoint in year 2004 (±95% CI in gray).

generate the observed spatial and temporal bistability, we in 2014 sampled fish, habitat characteristics (e.g. habitat-forming vegetation) and food (zooplankton) availability in 32 shallow bays along a 360 km stretch of the central coast, selected to form gradients in distance to the open sea and wave exposure (Supplementary Fig. 5). We sampled adult fish during spawning in spring (May), and young-of-the-year juveniles in late summer (August). Using path analysis to tease apart direct and indirect relationships[48], we then compared the fit of 14 multivariate hypotheses of the direct and indirect drivers of perch and stickleback recruitment, as graphical network models of interacting paths[36,49] (see "Methods" and Supplementary Table 2 for details). The best-fitting model (#12) fitted the data well ($p = 0.878$), and included several direct and indirect relationships that together demonstrate the pivotal role of predator–prey reversal for piscivore (perch and pike) recruitment (Supplementary Table 3, Fig. 4). High abundance of adult piscivores in spring had a strong negative influence on adult stickleback abundance; a negative predation effect supported by experiments[44], field surveys[36] and the frequent occurrence of stickleback remains in perch and pike stomachs[36]. Adult piscivore abundance also had an *indirect*, negative influence on stickleback recruitment (density of juvenile stickleback in summer); a temporally lagged relationship mediated by a positive, direct influence of adult stickleback abundance in spring on stickleback recruitment. High adult piscivore abundance in spring also positively influenced piscivore juvenile abundance in summer. However, this relationship was *indirect* and mediated by a strong negative influence of adult stickleback abundance in spring on piscivore recruitment (juvenile abundance in summer). This predator–prey reversal path is supported by experiments and field surveys[43,45] and was needed for the models to fit the data well (Supplementary Table 2). In addition to these predation effects, the % bottom cover of all benthic vegetation positively influenced adult stickleback abundance in spring[36], the cover of rooted vegetation positively influenced perch and pike juvenile abundance in summer[50], and high wave exposure positively influenced juvenile stickleback abundance in summer. In summary, the path analysis clearly suggests that bimodality in relative predator dominance at both adult and juvenile stages also in this smaller dataset (Supplementary Fig. 6) is partly self-sustained: perch and pike dominance in spring sustains their own recruitment by reducing stickleback predation on the earliest life stages, whereas stickleback dominance supports stickleback population development by suppressing predator recruitment.

## Discussion

To our knowledge this study is the first to describe a regime shift that spatially propagates through a marine ecosystem. Combined with previous studies on the stickleback increase, our findings highlight how environmental and biotic changes together facilitated the rise and spread of stickleback dominance: reduced predation pressure due to declining stocks of large predatory fish along the coast and in the open sea increased stickleback survival[34–37], while eutrophication (stimulating food production) and warming increased stickleback population growth[35,38,40,51,52]. In turn, predation by the hyper-abundant stickleback on benthic grazers and zooplankton generates system-wide trophic cascades that benefit fast-growing, filamentous algae[36,44,53] at the expense of habitat-forming benthic vegetation[36] and reduces resilience to eutrophication[44,54]. Our ecosystem field survey, supported by past experiments[43,45], also shows that stickleback suppresses perch and pike recruitment through predator–prey reversal (Fig. 4); a positive feedback that appears to reinforce the shift to stickleback dominance over time and space (Fig. 5). The reason(s) why this

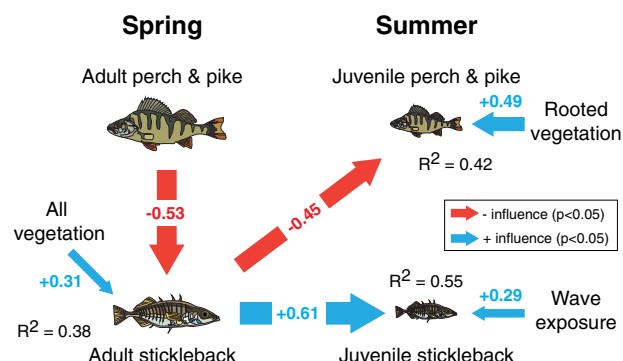

**Fig. 4 Predator–prey role reversal in shallow coastal bays.** Path diagram showing the best-fitting structural equation model (a.k.a. the "stickle-feed-back") of direct and indirect relationships between abundance of adult predators (perch and pike) and stickleback in spring, abundance of juvenile (young-of-the-year) predators (perch and pike) and stickleback in summer, seabed cover of aquatic vegetation and wave exposure, based on survey data from 31 shallow bays sampled in 2014. Red and blue arrows are significant ($p < 0.05$) negative and positive relationships, respectively. Arrow thickness is proportional to standardized path coefficients, also shown in colored texts.

vast shift has gone relatively unnoticed[55] could be its gradual nature, but also that the regular fish monitoring program excludes smaller-bodied fish like stickleback, and is concentrated to inner archipelago areas not (yet) reached by the stickleback front.

Whether this spatial regime shift constitutes a "critical transition" to a stable stickleback state is still an open question. On the one hand, our results give clear hints of a critical transition; (i) bimodality over time and space, even before the major stickleback increase (Fig. 2), (ii) sudden breaks in local time-series (Fig. 3), and (iii) a strong internal feedback mechanism through predator–prey reversal (Fig. 4). Moreover, the gradual expansion of stickleback dominance towards the mainland so far shows no transient "boom-bust" pattern or cyclicity typical of many consumer fronts[31]; a fact most likely explained by stickleback (i) spreading over an increasingly large area and (ii) being extreme generalists not limited by certain prey types[56], effectively reducing intraspecific competition associated with their dramatic population increase[57]. On the other hand, several of the likely drivers of the shift (e.g. seal and cormorant predation on large predatory fish, warming, etc.) have gradually increased over time along with stickleback abundance (Fig. 5). Therefore, demonstrating that stickleback dominance is persistent (i.e. upheld by internal feedbacks, and not by external conditions) would require reversing the driver(s) that caused the shift and then demonstrate that stickleback predation still restricts predator recovery; a hysteresis effect[5,6]. Reducing stickleback numbers may in theory improve recruitment of the highly local populations of perch and pike[34,43] and offer a test of persistence, but only if the stressors that caused the perch and pike decline are first reduced/removed. Moreover, the high connectivity of Baltic Sea stickleback populations[58], their yearly coastward migrations and the vast spatial scale of the shift (Fig. 2), indicate that such measures may have to be conducted at *very* large (e.g. whole-basin) scales to be effective.

We do not yet know whether the spatial shift to stickleback dominance can be halted, but theoretical models provide interesting predictions. First, spatial or "gradual" regime shifts themselves indicate that the potential for system-wide hysteresis is low[12,13,17]. Therefore, the front should in theory continue to expand towards the mainland coast, even though perch and pike juveniles may still escape predation in the most wave-sheltered mainland bays and freshwater tributaries[45]. Consequently, halting

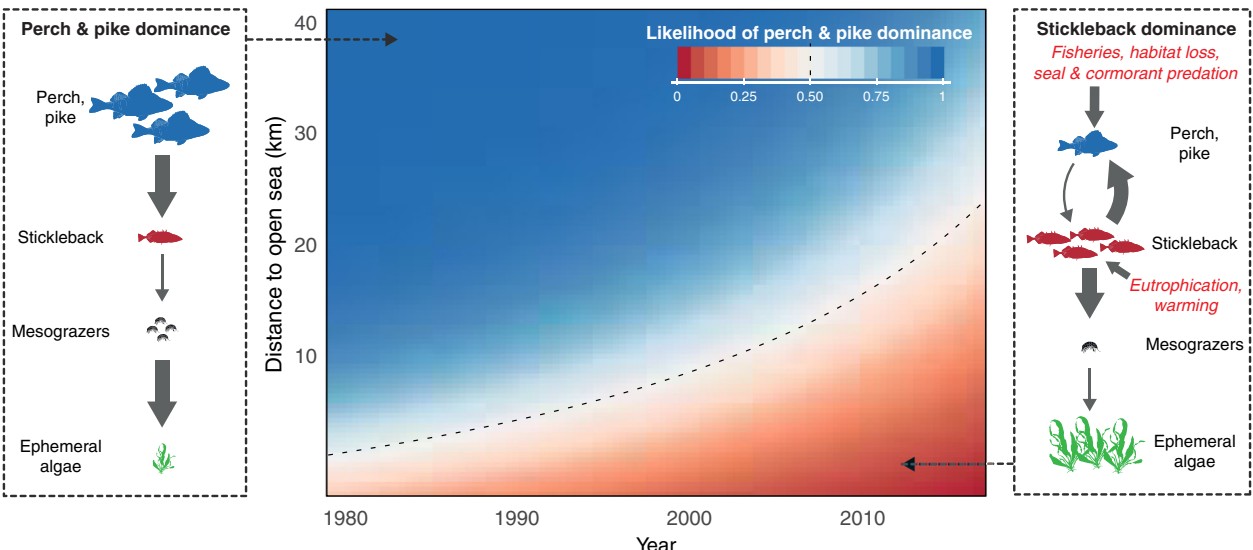

**Fig. 5 A spatial regime shift from perch and pike to stickleback dominance over ecosystem structure, processes and feedback mechanisms.** Center panel shows the influence of year and distance to open sea on the likelihood of perch and pike dominance (1/0; color gradient), after accounting for the influence of wave exposure and latitude. Black dashed line shows the 50% likelihood contour; the distance at which an individual bay is as likely to be perch- and pike- as stickleback-dominated. Left panel: empirically demonstrated four-level trophic cascade[36,44] in areas dominated by perch and/or pike (corresponding to blue areas in main panel). Right panel: conceptual model of how stickleback dominance (red areas in main panel) is favored by predator decline and environmental changes, and in turn favors ephemeral algae, by feeding on and controlling crustacean mesograzers[35,44]. Abundant stickleback also restrict perch and pike recruitment by feeding on their juvenile stages[34,43,45], which may indirectly benefit stickleback survival; a self-reinforcing feedback mechanism. Arrow thickness is proportional to the strength of interactions, and symbol numbers and sizes proportional to abundances. The mesograzer and algae symbols are courtesy of the Integration and Application Network, University of Maryland Center for Environmental Science (ian. umces.edu/symbols/).

or even reversing a spatially gradual regime shift may require unprecedented interventions that reverse underlying drivers at sufficiently large spatial and temporal scales[17]. This could in theory include (i) strengthening stocks of piscivorous fish that feed on stickleback (e.g. perch, pike, cod, large herring) through fisheries regulations along the coast and in the open sea[35], culling of fish-eating top predators like seals and cormorants[42], and restoring and protecting spawning and nursery areas of predatory fish, (ii) reducing stickleback densities through fisheries[34], and (iii) strengthening efforts to reduce eutrophication and climate change. Whether such actions could halt the stickleback front and even facilitate a reverse expansion of a perch and pike-dominated regime towards the open sea, increasing the many ecological, economic, and cultural values that these large predatory fish support, remains to be explored.

## Methods

**Spatial regime shifts over time and space**. To assess how the dominance of large predatory fish (Eurasian perch *Perca fluviatilis*, northern pike *Esox lucius*) and three-spined stickleback (*Gasterosteus aculeatus*) changed over time and space, we collated juvenile fish abundance data from 13073 samplings conducted during 39 years (1979–2017) in 486 bays along a 1200 km stretch of the Swedish Baltic Sea coast. We used juvenile fish surveys because they (i) include stickleback, (ii) were conducted along the entire coast, including the outer archipelago, and (iii) capture patterns of recruitment failure; a proposed driver of local perch and pike stock decline[34,35,37,45]. The samplings were conducted by various monitoring programs and research projects to quantify fish recruitment. Much of the data was extracted from the Swedish national database for coastal fish (http://www.slu.se/kul). Other fish species also occurred in the data, but we—like others[25,34,45]—focused on the most common and strongly interacting species.

The timing and placement of most of the fish surveys was not chosen with this study in mind. We therefore included data from as many surveys as possible, ensuring that the dataset covered (i) gradients in distance to open sea and wave exposure, (ii) the entire Swedish east coast, and (iii) the longest time period possible (Supplementary Fig. 7). Nearly all sampling (94%) was conducted during July-September. To achieve the best possible spatial coverage we also included some bays only sampled in October and (for one bay) June. Initial exploration of data from bays sampled monthly from June to October suggested there were no

large differences in the October and June data. Nine of the 486 bays occurred much further into the archipelago (49–67 km) than the rest (<41 km), resulting in very poor spatial coverage of the innermost half of the archipelago gradient. Moreover, these nine bays were all sampled after 2011, resulting in that space and time were confounded. Consequently, we excluded the nine bays (which were all predator-dominated) from further analyses.

The surveys were conducted in shallow coastal bays at $1.7 \pm 0.6$ depth (mean $\pm$ SD, range: 0.35–4.5 m, 99.9% $\leq 3.5$ m, N = 606). Juvenile fish were sampled using low-impact pressure waves; a standard method in the area[59]. In short, the ignition of a small, underwater explosive charge generates a pressure wave that stuns or kills all small fish with a swim bladder within the blast radius. Using the current Swedish standard (10 g Pentex explosive ignited by a 1 g non-electric charge), 2–20 cm fish are sampled within approximately a 5 m radius (ca. 80 m²). The fish are then collected using swing nets and snorkeling, identified and counted. All sampling was conducted by certified personnel and with required ethical permits.

The type and amount of explosives varied between surveys (1–25 g), and some surveys noted only floating (not sunken) fish. To account for these differences, we recalculated the total number of individual fish per species to the current detonation standard using experimentally derived conversion factors[59]. For surveys only reporting floating fish, we also calculated the expected total number of fish using the sunken:floating fish ratio from other surveys (N = 47–152 per species).

Since the method samples highly mobile organisms within a small surface area and very short time span, it is prone to high variability and false absences. We therefore averaged the species abundance of perch, pike and stickleback per bay and year (based on 5–177 samplings [detonations] per bay-year: mean $\pm$ SD = 15.6 $\pm$ 12.2). Most (75%) of the 477 included bays were sampled once, but 25% during multiple (2–34) years, generating 833 unique bay-year combinations. Since fish assemblages within a single bay can shift from perch and pike to stickleback dominance (for example, see Fig. 3), the 833 bay-years were treated as individual replicates. For each bay-year we then calculated the *relative predator dominance* (the summed abundance of perch and pike divided by the summed abundance of perch, pike and stickleback, ranging from 1 to 0 where 1 = complete perch and pike dominance, and 0 = complete stickleback dominance). We primarily used relative abundance because it captures the community state well[60–62] and reduces the "noise" in absolute abundance caused by the notoriously high year-to-year variability in fish recruitment[63]. Perch constituted 96% of the pooled abundance of perch and pike, but we included pike as well because they were occasionally more common than perch.

We also included data on four covariates known to influence juvenile fish abundance[36], when they were sampled: water depth (nearest 0.1 m, N = 606 bay-years), water surface temperature (nearest 0.1 °C, N = 598), salinity (nearest 0.1 psu, N = 300) and water visibility (Secchi depth to the nearest 0.1 m, estimated from turbidity, N = 325). Finally, we for each bay used GIS to calculate the distance

to open sea (m shortest water distance from the baseline) and the relative wave exposure ($m^2 s^{-1}$), based on a surface wave model (SWM) utilizing fetch and wind data averaged across 16 compass directions. The model mimics diffraction using empirically derived algorithms (for details, see ref. [64]).

**Temporal regime shift at Forsmark**. To test for a more classic temporal regime shift at the local (non-spatial) scale, we used the longest time-series from a single area; a 34-year data set from an annual monitoring program (1981–2017, no sampling in 1982, 1989, and 2003) conducted outside Forsmark (60.4°N, 18.2°E). The data was collected in an area used as a reference for estimations of effects of the release of heated cooling water from the Forsmark nuclear power plant. Only perch and stickleback (no pike) occurred.

**Importance of predator–prey reversal for fish recruitment**. To assess the relative importance of predation and predator–prey reversal for juvenile fish recruitment, we combined an ecosystem field survey with causal path analysis; a powerful approach to tease apart the role of direct and indirect interactions in ecosystems[36,49]. In 2014, we surveyed 32 shallow bays (<3.5 m depth) situated >10 km apart (or separated by natural fish movement barriers like deep water or land) along a 360 km stretch of the central Swedish Baltic Sea coast (Supplementary Fig. 5). Together, the bays formed gradients in distance to open sea and wave exposure, but also in perch, pike and stickleback abundance. This enabled us to statistically separate the influence of biotic interactions (e.g. predation) from abiotic conditions (e.g. wave exposure). In spring (May), after adult fish had migrated into the bays to spawn, we sampled the abundance of adult fish using 3–5 standard 30 m Nordic survey gill nets set overnight[36]. In 6–8 stations per bay we also quantified two environmental covariates known to influence fish: total % bottom cover of habitat-forming aquatic vegetation (visually estimated by snorkeling, separating rooted from non-rooted macrophytes; for details, see[36]) and density of zooplankton (juvenile fish food[37], estimated as the total plankton density per liter water). Zooplankton were sampled using a 25 cm diameter Epstein net (80um mesh), slowly pulled three times vertically from 0.7 m above the seabed to the water surface. Zooplankton were fixated in 5% formalin and then counted in the lab. In late summer (August) we at 6–8 stations per bay estimated fish recruitment (density of young-of-the-year fish, using low-impact pressure waves; see above) and % cover of habitat-forming benthic vegetation. The fish sampling was evaluated and approved by the ethical board on animal experiments of the County court of Uppsala, Sweden, permit C 139/13.

**Statistics and reproducibility**. *Spatial regime shifts over time and space*: we hypothesized that a shift from perch and pike to stickleback dominance started in wave-exposed bays in the outer archipelago, but then propagated towards the inner archipelago over time (Fig. 1a)—i.e. a statistical effect of space (distance to open sea), time (year) and, possibly, their interaction. Regime shifts have often been identified as breakpoints in time-series using non-parametric change-point detection methods. However, spatial regime shifts in heterogeneous systems should in theory be *gradual* (the mean of many small-scale shifts) at the whole-system scale[12,13]. Moreover, while multiple drivers could in theory influence whether bay assemblages are perch- or stickleback-dominated, change-point detection methods cannot handle multiple predictors and interactions between them. Finally, our response variable (relative predator dominance) is bounded between 0 and 1 and was bimodal, based on Hartigan's dip test of unimodality[65] with 10,000 permutations (Fig. 1b). Consequently, we used a binomial generalized linear model with a logit link function[66] to explore the effects of time (year), distance to open sea and their interaction on relatively predator dominance, while controlling for the influence of wave exposure (log-transformed) and latitude. After assessing assumptions of homoscedasticity by plotting deviance residuals vs. observed values for each predictor (Supplementary Fig. 8), and multicollinearity using the variance inflation factor (VIF; most <2, all <5), we identified the most parsimonious model by (i) comparing candidate models using Akaike's Information Criterion (AIC) and (ii) stepwise removal of non-significant terms (at α = 0.05). All statistical analyses were conducted using R v. 3.6.0[67].

Since 75% of the bays were sampled only once we could not include "bay" as a random factor. To assess the robustness of our results given the repeated sampling in 25% of the bays, we (i) generated a smaller dataset including one randomly chosen observation (year) per bay (N = 477), (ii) refitted the best binomial model using this new dataset, (iii) extracted the estimates, standard deviations and standard errors for each parameter, (iv) repeated the whole procedure 500 times, and iv) summarized the average results (Supplementary Table 4). The additive effects of time, distance to open sea, wave exposure and latitude remained, meaning that the conclusion of a spatial regime shift were robust. The interaction between time and distance was only significant in 29% of the runs; a discrepancy most likely explained by the lower power of a test based on a smaller sample size (N = 477 vs. 833 in the full dataset), as well as the influence of randomly including more perch- and pike- or stickleback-dominated years from the 25% (131) bays sampled >1 year.

To test whether the *maximum* spatial extent of stickleback dominance along the archipelago gradient increased over time, we first selected all stickleback-dominated bay-years (relative predator dominance ≤0.1, i.e. ≥90% stickleback). For each sampling year we then extracted the maximum distance from open sea,

separating wave-sheltered vs. -exposed bays (using a cutoff of $log10(m^2 s^{-1})$ >4). Since no bays were sampled >20 km from the open sea prior to 1995, we only used data from 1995–2017. Finally, we used a general linear multiple regression model to test how sampling year, wave exposure (two levels) and their interaction explained the maximum distance.

The bimodality in relative predator dominance seen over time and space (Figs. 1–3) could in theory be caused by variability in local abiotic conditions, and not by predator–prey interactions. We therefore tested whether any of four abiotic conditions estimated locally—water surface temperature, salinity, turbidity, water depth—could explain the variability in deviance residuals from the binomial glm (Supplementary Fig. 8), using a regular linear model (after assessing model assumptions and ruling out multicollinearity; see above). Salinity and turbidity (but not depth and temperature) had statistically significant but weak influences ($R^2$ = 0.11), and the bimodality clearly remained (Supplementary Fig. 9).

Finally, we explored how time, distance to open sea, wave exposure, latitude and their two- and three-way interactions influenced log-transformed abundances of (i) predators (perch and pike pooled) and (ii) stickleback, using general linear models. We identified the most parsimonious models as outlined above.

*Temporal regime shift at Forsmark*: using the 34-year Forsmark time series, we first tested for temporal breakpoints in logit-transformed relative predator dominance data using change-point detection (*strucchange*) for linear models[47]. This method estimates the optimal number and (if identified) position of breakpoints using the Bayesian Information Criterion (BIC). Second, we explored what temporal change(s) in perch and stickleback abundances that preceded the shift. Because of highly non-linear patterns we modeled the temporal changes using generalized additive models (GAM) as implemented in the *mgvc* package[68].

*Importance of predator–prey reversal for fish recruitment*: to assess the relative importance of predator–prey reversal for perch, pike and stickleback recruitment, we used statistical model selection based on path analyses; a form of structural equation modeling that can be used to tease apart direct vs. indirect (mediated) relationships between multiple (>2) variables, and thereby assess the relative importance of direct vs. indirect relationships in systems[48,69]. Initial data exploration using multiple regression showed that one bay was a clear outlier due to 0 juvenile perch and pike, generating (i) too high leverage (influence on statistical relationships), (ii) heteroscedasticity and (iii) non-normally distributed errors. Since we suspected that juveniles had already migrated out of this bay, the bay was excluded (resulting in N = 31). Removing this statistical outlier resulted in that the model fulfilled test assumptions and the overall fit more than doubled (adjusted $R^2$ increased from 0.17 to 0.37).

Based on ecological knowledge of the study system, we expressed 14 multivariate hypotheses of the direct and indirect drivers of perch and stickleback recruitment, as graphical network models of interacting paths[36,49]. Due to the relatively low sample size we restricted the number of paths to 7. The two simplest models assumed that perch+pike and stickleback juvenile abundance in summer (i.e. recruitment) was influenced by adult abundance in spring (stock recruitment) and cumulative cover of rooted vegetation, while adult abundance was explained by spring cumulative cover of all vegetation species[36,46], and distance to open sea or wave exposure[46]. The more complex models included combinations of known predator–prey interactions: perch and pike controlling adult stickleback in spring through predation[36], stickleback feeding on juvenile perch and pike[45], and stickleback competing with juvenile perch for zooplankton prey[37]. We then analyzed each model using piecewise path analysis as implemented in the *piecewiseSEM* package[48]. First, we tested the goodness of fit of each model to the data using Shipley's test of directional separation (D-sep)[70]. If missing paths were identified, they were included in a new model. For the models that fitted the data (p > 0.05) we used the Akaike's Information Criterion corrected for small samples (AICc) (calculated using Shipley's general approach to calculate AIC for path analysis[71]) to compare relative model fit. A summary of all candidate models and details of the best-fitting model are presented in Supplementary Tables 1 and 2, respectively. The strength of paths in the best-fitting model are presented using standardized path coefficients, which (based on the best-fitting model, see Fig. 3) mean that 1 SD increase in pooled adult perch and pike abundance reduces adult stickleback abundance by 0.53 SD. We also calculated the amount of variation ($R^2$) in adult stickleback abundance, pooled juvenile perch and pike, and juvenile stickleback abundance, that was explained by the paths. Finally, we tested whether the relative predator dominance of adult and juvenile fish in this smaller dataset (N = 31) was also uni- or bimodal, using Hartigan's dip test (for details, see above).

**Reporting summary**. Further information on research design is available in the Nature Research Reporting Summary linked to this article.

## Data availability
All data generated or analyzed during this study are included in this published article (and its Supplementary Information files). The source data underlying the large-scale statistical analyses and plots shown in Figs. 1–3 and 5 are provided in Supplementary Data 1. The source data from the 2014 ecosystem survey, underlying the analysis and plot shown in Fig. 4, is provided in Supplementary Data 2. Much of the fish survey data was extracted from the Swedish national database for coastal fish (for more information, see http://www.slu.se/kul).

## Code availability

Standard functions in the R environment (no custom code) were used to generate the statistical analyses and figures.

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

## Acknowledgements

This study is a product of project *PlantFish*, funded by the Swedish Research Council Formas (2013-1074), HM Carl XVI Gustaf's Foundation for Science and Education (2014-0002), the Baltic Sea 2020 foundation, the Stockholm University Baltic Sea Centre (Askö grants) and in-kind support from Stockholm University, the Swedish Agricultural University (SLU) and Groningen University. The historical surveys of juvenile fish were conducted by many monitoring programs and research projects, funded by various national agencies and national/international research councils including the Swedish research councils VR and Formas, the Swedish Agency for Marine and Water Management, the Swedish Environmental Protection Agency, the former Swedish Board of Fisheries, the Swedish County administrative boards, the Uppland Foundation, EU Interreg (IIIA and IIIB) and Forsmarks kraftgrupp. We humbly acknowledge all the resources and hard work that went into collecting the data, and sincerely thank those individuals, groups, organizations and agencies that willingly shared it with us. We would particularly like to acknowledge (in alphabetic order) G. Johansson, P. Karås, L. Ljunggren, J. Persson and A. Sandström, who conducted many of the surveys. We also thank Å. Austin, P. Jacobson, G. Johansson, G. Lilliesköld-Sjöö, L. Lozys, E. Mörk, M. van Regteren, J. Sagerman, S. Skoglund, M. van der Snoek and V. Thunell for assistance with the ecosystem field survey in 2014. Finally, we thank R. Elmgren, L. Gamfeldt and M. Casini for helpful comments on earlier versions of the article. Open access funding provided by Stockholm University.

## Author contributions

J.S.E., U.B., B.K.E., and G.S. conceived the study; G.S., U.B., and M.E. collated the large juvenile fish dataset; J.S.E., G.S., U.B., B.K.E., J.P.H., and S.D. conducted the ecosystem field survey in 2014; JSE analyzed the data with support from G.S., B.K.E., and S.D.; J.S.E., B.K.E., and U.B. led the writing of the manuscript; G.S., J.P.H., S.D., and M.E. reviewed previous versions.

## Competing interests
The authors declare no competing interests.
