## [Peer Review File · Communications Biology]

Reviewers' comments:

Reviewer #1 (Remarks to the Author):

This is an important and impressive paper documenting how ecological regime shifts could propagate through space, and moreover, it provides compelling evidence for the ecological mechanisms behind the observed changes. The spatial component of regime shifts is often overlooked but might be important because asynchronous local shifts might result in a gradual global change. Such gradual changes could easily be interpreted as a linear and a reversible response to external perturbations, while the system in fact has gone through many local regime shifts that are locked by irreversible positive feedbacks. The paper should therefore be of interest to a wider field of researchers.

Human removal of large predatory fishes has initiated numerous regime shifts in aquatic ecosystems, and "predator-prey reversal" has been suggested to be one important positive feedback mechanism responsible for shifting systems from a predator to a prey dominated state. In the present paper, Eklöf and co-workers present a large spatio-temporal dataset from the Baltic Sea. Their analyses show that during the last 40 years, regime shifts have propagated along an environmental gradient from the outer towards the inner coastal areas. The shifts involve a change from a system dominated by large predatory fishes to a system dominated by small prey fishes. To investigate the mechanisms behind the shifts, they conducted an extensive tailored survey and show through innovative path analyses that predator-prey reversal is likely to be the key mechanism behind the observed change.

I find the statistical methods applied to be appropriate and the results convincing. Figures are carefully prepared and informative. I have only some minor comments. The Result and Method chapters are long and could be shortened and simplified to improve readability. Specifically, I would suggest that the analyses of local co-variables, analyses of abundances and possible effects of random factors (results lines 173-188 and methods lines 360-385) are moved to the Supplementary Material. Finally, I would like to see measures for the overall model fit of the GLMs in Table S1.

Reviewer #2 (Remarks to the Author):

The paper demonstrates a spatial regime shift from a large predatory fish species to a small prey fish in the Baltic Sea. The authors demonstrate that this shift slowly propagated from wave-exposed to wave-sheltered areas eventually changing the fish community in a vast area. The authors made use of an impressive empirical longer-term data set and sampled an additional data set to demonstrate the positive feedback process that facilitated the regime shift, i.e. a predator-prey role reversal from large predators controlling prey fish to more abundant prey fish now controlling predators by preying on their juvenile stages.

To my opinion the findings of the paper are novel since they validated model-derived insights in how regime shifts can develop spatially and how they are stabilized by a feedback loop. This has been seldomly if ever shown in such a rigorous way. Hence the study is unique and also relevant as a template for better understanding such shifts in other areas and ecosystems.

The work is furthermore very convincing since it reliably puts observed patterns in a theoretical framework. The work will likely influence thinking in the field, motivating studies that better link temporal and spatial regime shift dynamics. Such studies would facilitate a better understanding of

the processes that induced and stabilize catastrophic changes also in other ecosystem types. Moreover, the study uses statistical methods in a very rigorous way. It combines GLMs with Structural Equation Modelling, especially the latter being very suitable of showing the many interactions in a real ecosystem.

Overall, I would congratulate the authors to this study since it is a great (and rare) example of how ecological theory is tested in a real ecosystem. The study further demonstrates the value of field sampling and monitoring over large temporal and spatial scales, since only after a longer time and large areas the patterns demonstrated here will be visible.

Reviewer #3 (Remarks to the Author):

In this paper, the authors employ a large data set to describe the food web consequences of removing high trophic level fish (pike and perch) from a coastal marine ecosystem. Their data show a switch to dominance by a lower-trophic level fish (stickleback) that are prey to the pike and perch. This switch to stickleback dominance proceeds from the outer archipelago, which is less favorable habitat for pike and perch, towards the inner archipelago, which is more favorable habitat for those species. The authors demonstrate changing trophic linkages after the removal of the higher trophic level fish, in particular a predator-prey reversal, whereby the sticklebacks, after release from predation, exert top-down control on the early life history stages of pike and perch. The authors also demonstrate a high degree of bimodality in their data, with different sampling events revealing either strong numerical dominance by perch/pike or by sticklebacks, and they make the inference that trophic feedbacks tend to maintain the community in either state.

These are very interesting results. While neither an increase in a lower trophic level from top-down control nor predator-prey reversal is novel in the literature as a consequence of the removal of a higher trophic level fish from an ecosystem, this is a well-documented example that would make a valuable addition to the literature on the ecological consequences of overfishing.

However, the central claim that these data are an example of a spatial regime shift occurring over a large area in a coastal ecosystem is not supported adequately. This is a combination of a failure to be explicit both about defining what a regime shift is and defining the evidence that is needed to demonstrate that a regime shift has occurred, as well as the difficulty inherent in demonstrating the existence of theoretically predicted types of regime shifts with observational data. My specific concerns are outlined below:

1) What is a regime shift? This central term is not explicitly defined in the paper. This is a common problem in the regime shift literature. The Introduction of this paper initially seems to imply that any release of lower trophic levels after the removal of higher trophic level fish qualifies as a regime shift (lines 33-35). But the Introduction also seems to be arguing that regime shifts involve alternative states (e.g., line 38). A variety of papers employing these two different criteria for defining regime shifts are cited. Some define regime shifts as any abrupt, persistent shift in a complex system (e.g.; deYoung et al., 2008). The trouble is that this definition is so broad as to be fairly useless. Abrupt, persistent changes occur in nature all the time, for any number of reasons, and calling any such instance a regime shift is to lump together any number of different dynamics. Di Lorenzo and Ohman (2013), for instance, very elegantly demonstrated how purely random variability can produce abrupt, persistent shifts between contrasting ecosystem states. Other references in the paper more explicitly invoke some aspects of alternative stable states (e.g.; Gardmark et al., 2015; Rocha et al., 2018). In

any case, a precise definition is needed in order to properly frame the central argument of the paper.

2) In a similar vein, and flowing from the first comment, the paper needs to explicitly lay out the evidence that is needed to demonstrate the presence of a regime shift. Absent a specific definition of what is being tested for, and a specific exposition of the evidence that would be needed to satisfy that test, the paper lacks sufficient rigor.

3) Additionally, the evidence that is presented is either consistent with simple trophic dynamics that do not require a "regime shift" to explain, or fall short of demonstrating alternative states.

a) The presence of a bimodal distribution (Fig. 1) is not evidence of alternative states. A number of mechanisms can create bimodal distributions in ecological data, including integration of white noise variability (Di Lorenzo & Ohman, 2013), and phase transitions, which are fully reversible transitions (Dudgeon et al., 2010). In recognition of the fact that bimodality is necessary, but not sufficient, evidence for demonstrating alternative states, it has been proposed that bimodality in observational data should be used as a "hint" that theoretically-predicted regime shifts are present in a system (Scheffer & Carpenter, 2003). The finding of bimodality is an important result, but treating this as evidence that a regime shift (involving alternative states) has occurred is over-interpreting that result.

b) The progression of stickleback-dominated bays from offshore to onshore areas (Fig. 2) appears to be consistent with density-dependent patterns of habitat occupancy by perch and pike, involving the utilization of lower quality offshore habitats at high densities, and abandonment of these areas at lower densities. This, and simple trophic effects on stickleback abundance, appear sufficient to explain the offshore-onshore progression of stickleback dominance, without invoking any complex regime shift dynamics. As the authors note, the bimodal distribution around the regression lines in Fig. 2 (and similar patterns in Figs. S1 and S4) is intriguing. The demonstration of predator-prey reversal in this paper is an appealing explanation as a possible mechanism explaining this bimodality. However, to really demonstrate that predator-prey reversal does maintain the prey-rich state, perturbation to the perch and pike would have to be reversed. Then the central question of whether stickleback dominance persisted even after the perturbation to their predators was removed could be answered. This would provide a meaningful test for alternative states (more on this below). Without this level of evidence, the bimodality and predator-prey reversal is intriguing and important, but not as novel as a conclusive demonstration of alternative states would be.

c) Fig. 3 simply shows that when the predatory fish is removed, the biomass of sticklebacks increases rapidly. This is hardly surprising, and does not require the invocation of any complex regime shift dynamics to explain.

d) Fig. 4 presents the path analysis supporting the interpretation of predator-prey reversal. This result is very interesting and a valuable contribution. However, another step is required to show that these dynamics maintain stickleback dominance – the pressure on pike and perch (fishing, predation, habitat degradation) would have to be reversed to see if the system switched back to the predator-rich state (a reversible phase transition and not alternative states) or did not. Even in the latter case, where a prey-rich state at first appears to be a self-sustaining alternative state that persists after the cessation of pressure on the predators, it may simply be that the prey-rich state proves to be a transient phenomenon (not a regime shift), and the system reverts to the original configuration after a time lag (Frank et al., 2011).

e) Fig. 5 outlines the putative predator-prey reversal mechanism, along with a trophic cascade. The system is presented as consisting of four trophic levels. However, as noted above, the perturbation that triggered the transition to stickleback dominance came from the fifth trophic level

(humans/birds/seals). To really understand whether alternative states operate in this system, observations at contrasting levels of top-down control by this fifth trophic level would be needed. I recognize that these data are likely not available, but the authors should be explicit about that absence, and cautionary that demonstration of alternative states (if that is what is meant by a regime shift) would at the least require comparison across varying pressures in trophic control on perch/pike.

This paper will make a very valuable contribution, but the central claims are poorly defined and not supported by the data. The paper should be rewritten to focus on the demonstration of predator-prey reversal, and should be much more cautionary about interpreting the evidence as demonstrating a large-scale regime shift.

Finally, the data availability statement ("All data is available upon request from the corresponding author") is really not up to the community standard any more. Data and code for reproducing the results should be posted in a publicly available repository, and Communications Biology should make this a requirement for publication. This is simply the direction that science is going, and it is a very positive development in terms of guaranteeing reproducibility.

deYoung, B., M. Barange, G. Beaugrand, R. Harris, R. I. Perry, M. Scheffer, and F. Werner. 2008. Regime shifts in marine ecosystems: detection, prediction and management. *Trends in Ecology & Evolution* 23:402–409.

Dudgeon, S. R., R. B. Aronson, J. F. Bruno, and W. F. Precht. 2010. Phase shifts and stable states on coral reefs. *Marine Ecology Progress Series* 413:201–216.

Frank, K. T., B. Petrie, J. A. D. Fisher, and W. C. Leggett. 2011. Transient dynamics of an altered large marine ecosystem. *Nature* 477:86–89.

Gardmark, A., M. Casini, M. Huss, A. van Leeuwen, J. Hjelm, L. Persson, and A. M. de Roos. 2015. Regime shifts in exploited marine food webs: detecting mechanisms underlying alternative stable states using size-structured community dynamics theory. *Philosophical Transactions of the Royal Society B-Biological Sciences* 370.

Di Lorenzo, E., and M. D. Ohman. 2013. A double-integration hypothesis to explain ocean ecosystem response to climate forcing. *Proceedings of the National Academy of Sciences* 110:2496–2499.

Rocha, J. C., G. Peterson, O. Bodin, and S. Levin. 2018. Cascading regime shifts within and across scales. *SCIENCE* 362:1379+.

Scheffer, M., and S. Carpenter. 2003. Catastrophic regime shifts in ecosystems: linking theory to observations. *Trends in Ecology & Evolution* 18:648–656.

Point-by-point response to reviewers

Reviewers' comments:

Reviewer #1 (Remarks to the Author):

This is a an important and impressive paper documenting how ecological regime shifts could propagate through space, and moreover, it provides compelling evidence for the ecological mechanisms behind the observed changes. The spatial component of regime shifts is often overlooked but might be important because asynchronous local shifts might result in a gradual global change. Such gradual changes could easily be interpreted as a linear and a reversible response to external perturbations, while the system in fact has gone through many local regime shifts that are locked by irreversible positive feedbacks. The paper should therefore be of interest to a wider field of researchers.

Human removal of large predatory fishes has initiated numerous regime shifts in aquatic ecosystems, and "predator-prey reversal" has been suggested to be one important positive feedback mechanism responsible for shifting systems from a predator to a prey dominated state. In the present paper, Eklöf and co-workers present a large spatio-temporal dataset from the Baltic Sea. Their analyses show that during the last 40 years, regime shifts have propagated along an environmental gradient from the outer towards the inner coastal areas. The shifts involve a change from a system dominated by large predatory fishes to a system dominated by small prey fishes. To investigate the mechanisms behind the shifts, they conducted an extensive tailored survey and show through innovative path analyses that predator-prey reversal is likely to be the key mechanism behind the observed change.

Answer 1: We sincerely thank the reviewer for these positive and kind words. We agree this is an important study, both from a fundamental and applied perspective.

I find the statistical methods applied to be appropriate and the results convincing. Figures are carefully prepared and informative. I have only some minor comments. The Result and Method chapters are long and could be shortened and simplified to improve readability. Specifically, I would suggest that the analyses of local co-variables, analyses of abundances and possible effects of random factors (results lines 173-188 and methods lines 360-385) are moved to the Supplementary Material.

Answer 2: We agree the Results section can be shortened. We have therefore moved the analysis of local co-variables to the Methods section, which has no word limit (see lines 772-786). We wish to keep the analysis of the possible effects of random factors in the Methods section (see lines 762-763) and not in a supplement, because i) the analysis is important for readers given that our main analyses does not include random factors, and ii) the Methods section has no word limit. Finally, we suggest to keep the analyses of fish abundances in the main text (lines 354-370), for three reasons:

i) A temporal or spatial change in the relative dominance 'index' could in theory be caused by a change in only stickleback or only perch/pike abundances, whereas the theory of a regime shift entails that stickleback increase while perch/pike decline. This is now explained in the Results section (lines 354-357). "Relative predator dominance is a ratio that in theory could respond to changes in only stickleback- or perch and pike numbers. Therefore, we also tested how the respective abundances of perch, pike and stickleback changed over time and space, using linear models (see Methods for details)."

ii) Also the timing of the changes in abundances of predator and prey is important, to indicate to what extent the stickleback increase is caused by the perch and pike decline.

iii) The general (coast-wide) decline in perch and pike recruitment over time is in itself an important and novel finding, that may get lost if moved to a Supplement.

However, if the Editor thinks that moving these sections to the Supplements is necessary to be able to accept the paper, we will swiftly do so.

Finally, I would like to see measures for the overall model fit of the GLMs in Table S1.

Answer 3. We have now added R^2 values to Table S1 (see lines 1374-1378). The relatively low levels (R^2 : 0.17-0.25) are explained by the strong bimodality (see main Figures 1-2 and Supplementary Figure S4).

Reviewer #2 (Remarks to the Author):

The paper demonstrates a spatial regime shift from a large predatory fish species to a small prey fish in the Baltic Sea. The authors demonstrate that this shift slowly propagated from wave-exposed to wave-sheltered areas eventually changing the fish community in a vast area. The authors made use of an impressive empirical longer-term data set and sampled an additional data set to demonstrate the positive feedback process that facilitated the regime shift, i.e. a predator-prey role reversal from large predators controlling prey fish to more abundant prey fish now controlling predators by preying on their juvenile stages.

To my opinion the findings of the paper are novel since they validated model-derived insights in how regime shifts can develop spatially and how they are stabilized by a feedback loop. This has been seldomly if ever shown in such a rigorous way. Hence the study is unique and also relevant as a template for better understanding such shifts in other areas and ecosystems.

The work is furthermore very convincing since it reliably puts observed patterns in a theoretical framework. The work will likely influence thinking in the field, motivating studies that better link temporal and spatial regime shift dynamics. Such studies would facilitate a better understanding of the processes that induced and stabilize catastrophic changes also in other ecosystem types. Moreover, the study uses statistical methods in a very rigorous way. It combines GLMs with Structural Equation Modelling, especially the latter being very suitable of showing the many interactions in a real ecosystem.

Overall, I would congratulate the authors to this study since it is a great (and rare) example of how ecological theory is tested in a real ecosystem. The study further demonstrates the value of field sampling and monitoring over large temporal and spatial scales, since only after a longer time and large areas the patterns demonstrated here will be visible.

Answer 4: We sincerely thank the reviewer for the overwhelmingly positive comments.

Reviewer #3 (Remarks to the Author):

In this paper, the authors employ a large data set to describe the food web consequences of removing high trophic level fish (pike and perch) from a coastal marine ecosystem. Their data show a switch to dominance by a lower-trophic level fish (stickleback) that are prey to the pike and perch. This switch to stickleback dominance proceeds from the outer archipelago, which is less favorable

habitat for pike and perch, towards the inner archipelago, which is more favorable habitat for those species. The authors demonstrate changing trophic linkages after the removal of the higher trophic level fish, in particular a predator-prey reversal, whereby the sticklebacks, after release from predation, exert top-down control on the early life history stages of pike and perch. The authors also demonstrate a high degree of bimodality in their data, with different sampling events revealing either strong numerical dominance by perch/pike or by sticklebacks, and they make the inference that trophic feedbacks tend to maintain the community in either state.

These are very interesting results. While neither an increase in a lower trophic level from top-down control nor predator-prey reversal is novel in the literature as a consequence of the removal of a higher trophic level fish from an ecosystem, this is a well-documented example that would make a valuable addition to the literature on the ecological consequences of overfishing.

Answer 5. We thank the reviewer for this positive note; we also think the results are very interesting and hope that the manuscript will make a valuable addition to the literature.

However, the central claim that these data are an example of a spatial regime shift occurring over a large area in a coastal ecosystem is not supported adequately. This is a combination of a failure to be explicit both about defining what a regime shift is and defining the evidence that is needed to demonstrate that a regime shift has occurred, as well as the difficulty inherent in demonstrating the existence of theoretically predicted types of regime shifts with observational data. My specific concerns are outlined below:

1) What is a regime shift? This central term is not explicitly defined in the paper. This is a common problem in the regime shift literature. The Introduction of this paper initially seems to imply that any release of lower trophic levels after the removal of higher trophic level fish qualifies as a regime shift (lines 33-35). But the Introduction also seems to be arguing that regime shifts involve alternative states (e.g., line 38). A variety of papers employing these two different criteria for defining regime shifts are cited. Some define regime shifts as any abrupt, persistent shift in a complex system (e.g.; deYoung et al., 2008). The trouble is that this definition is so broad as to be fairly useless. Abrupt, persistent changes occur in nature all the time, for any number of reasons, and calling any such instance a regime shift is to lump together any number of different dynamics. Di Lorenzo and Ohman (2013), for instance, very elegantly demonstrated how purely random variability can produce abrupt, persistent shifts between contrasting ecosystem states. Other references in the paper more explicitly invoke some aspects of alternative stable states (e.g.; Gardmark et al., 2015; Rocha et al., 2018). In any case, a precise definition is needed in order to properly frame the central argument of the paper.

Answer 6: We thank the reviewer for this important critique (see also related questions/comments below). We agree we should have been more clear with definitions, particularly since there is no single definition of a regime shift (see below). We have now added a new 1st paragraph to the Introduction, that defines the concepts and explains their history (see lines 40-6).

In line with other marine scholars before us (deYoung et al. 2008, Andersen et al. 2009, Conversi et al. 2014), we in this paragraph define regime shifts as abrupt and long-term shifts in the structure and function of a complex system, that include (but are not exclusive to) shifts between stable states. This definition is based on empirical evidence rather than the theory of catastrophic/critical transitions, as discussed by Conversi et al. (Conversi et al. 2014). We use this definition for two main reasons:

1) to our understanding, the term 'regime shift' in ecology was coined in *marine science* in the mid 1990s to describe *environmentally-driven*, rapid changes in biological communities (Collie et al. 2004) – i.e. phase shifts, using the term mentioned by the reviewer. In the early 2000s the term was

gradually adopted in the broader ecological literature on ecosystems shifts, but then sometimes with a new definition, equating it with 'critical transitions' = shifts between stable states (Scheffer & Carpenter 2003). The reason for using the term 'regime' instead of 'state' was that "...it might be more appropriate to use words such as 'regimes' or 'attractors' instead of terms such as 'stable states' or 'equilibria' that seem to exclude dynamics" (Scheffer & Carpenter 2003).

2) A definition that stems from empirical observation, and includes – but is not restricted to) catastrophic shifts between stable states – is much more practical for marine management purposes, and can be used for various habitats/ecosystems (e.g. both benthic and pelagic shifts in the ocean), even in cases where the link to the mathematical theory is not yet established (Conversi et al. 2014).

The lack of a single definition of regime shifts – which has been acknowledged by others (Conversi et al. 2014) – is problematic but not uncommon. For example, the related concept 'resilience' has been defined in >50 different ways. We agree with the reviewer that defining the term is therefore an important step to increase clarity. However, as problematic is that many authors who explicitly equate regime shifts with critical transitions *also* often fail to show that the shifts are indeed persistent. We raise this issue because reviewer #3 correctly points out that our data does not show whether stickleback dominance is persistent, but then refers to (Rocha et al. 2018) as an example of studies "...that more explicitly invoke some aspects of alternative stable states". A quick scrutiny of the 'Regime shift database' used by Rocha et al. (www.regimeshifts.org) reveals that most of the cases are also *not* (yet) demonstrated to be 'critical transitions', and that many are likely to be mere 'phase shifts'. Undoubtedly, there are many similar examples in the literature, illustrating the complexity of this issue.

2) In a similar vein, and flowing from the first comment, the paper needs to explicitly lay out the evidence that is needed to demonstrate the presence of a regime shift. Absent a specific definition of what is being tested for, and a specific exposition of the evidence that would be needed to satisfy that test, the paper lacks sufficient rigor.

Answer 7: We agree and have therefore in the Discussion now clarified what is needed to demonstrate whether a regime shift is a 'critical transition' or not, in a heavily revised 2nd paragraph (see lines 373-377). We explain that our results give hints – not proof - of a critical transition, and that manipulation of underlying drivers and/or stickleback would be needed to test for persistence.

3) Additionally, the evidence that is presented is either consistent with simple trophic dynamics that do not require a "regime shift" to explain, or fall short of demonstrating alternative states.

Answer 8: The evidence of spatial and temporal bimodality, and the fact that predator-prey reversal has a stronger effect on fish recruitment than abiotic drivers, are clear hints of a critical transition. But without large-scale, long-term manipulation of drivers and/or stickleback abundances, we cannot rule out that this is only a 'simple' case of mesopredator release. This is now clearly outlined in the Discussion new paragraph #2 (lines 371-376).

a) The presence of a bimodal distribution (Fig. 1) is not evidence of alternative states. A number of mechanisms can create bimodal distributions in ecological data, including integration of white noise variability (Di Lorenzo & Ohman, 2013), and phase transitions, which are fully reversible transitions (Dudgeon et al., 2010). In recognition of the fact that bimodality is necessary, but not sufficient, evidence for demonstrating alternative states, it has been proposed that bimodality in observational data should be used as a "hint" that theoretically-predicted regime shifts are present in a system

(Scheffer & Carpenter, 2003). The finding of bimodality is an important result, but treating this as evidence that a regime shift (involving alternative states) has occurred is over-interpreting that result.

Answer 9: We agree that we should have been more clear about what our results show, what they hint at, and what they don't show. We now clarify that we present patterns in field data that are consistent with a regime shift (in the broad sense), but that proving that predator- vs. prey-dominance are alternative stable states would require manipulation of environmental drivers/and or stickleback at unprecedented scales (see lines 430-439): "... On the other hand, several of the likely drivers of the shift (e.g. seal and cormorant predation, warming, etc.) have gradually increased over time along with stickleback abundance (Fig. 5). Therefore, demonstrating that stickleback dominance is persistent (i.e. upheld by internal feedbacks, and not by external conditions) would require reversing the driver(s) that caused the shift and then demonstrate that stickleback predation still restricts predator recovery ; a hysteresis. Reducing stickleback numbers may in theory improve recruitment of the highly local populations of perch and pike and offer a test of persistence, but only if the stressors that caused the perch and pike decline are first reduced/removed."

b) The progression of stickleback-dominated bays from offshore to onshore areas (Fig. 2) appears to be consistent with density-dependent patterns of habitat occupancy by perch and pike, involving the utilization of lower quality offshore habitats at high densities, and abandonment of these areas at lower densities. This, and simple trophic effects on stickleback abundance, appear sufficient to explain the offshore-onshore progression of stickleback dominance, without invoking any complex regime shift dynamics. As the authors note, the bimodal distribution around the regression lines in Fig. 2 (and similar patterns in Figs. S1 and S4) is intriguing. The demonstration of predator-prey reversal in this paper is an appealing explanation as a possible mechanism explaining this bimodality. However, to really demonstrate that predator-prey reversal does maintain the prey-rich state, perturbation to the perch and pike would have to be reversed. Then the central question of whether stickleback dominance persisted even after the perturbation to their predators was removed could be answered. This would provide a meaningful test for alternative states (more on this below). Without this level of evidence, the bimodality and predator-prey reversal is intriguing and important, but not as novel as a conclusive demonstration of alternative states would be.

Answer 10: As described in the preceding answers, we agree that proving that this regime shift is a critical transition would require a test of the persistence of stickleback – by removing/reducing other perturbations on perch and pike (and expect a hysteresis), or by perturbing stickleback and expecting a shift, *if* the system is in the bimodal range of conditions. We have now added this reasoning to the Discussion (lines 471-510).

c) Fig. 3 simply shows that when the predatory fish is removed, the biomass of sticklebacks increases rapidly. This is hardly surprising, and does not require the invocation of any complex regime shift dynamics to explain.

Answer 11: We agree that Fig 3 supports the idea that when predators decline, stickleback increase rapidly. Even though this finding may seem trivial, this result has not been shown before in the case of the stickleback increase, and therefore adds an important piece of information. Moreover, the local breakpoint shows that even though the large-scale (regional) shift occurs gradually (Fig. 2), the *local* (bay-level) shift from perch- to stickleback dominance can be abrupt – as predicted by theory on spatial or 'gradual' regime shifts (van Nes & Scheffer 2005, Bel et al. 2012). These observations are, again, not *proof* of a critical transition, but constitute yet another important condition for a spatial regime shift. We have now modified the Results sections to better explain this reasoning (lines 342-345).

d) Fig. 4 presents the path analysis supporting the interpretation of predator-prey reversal. This result is very interesting and a valuable contribution. However, another step is required to show that these dynamics maintain stickleback dominance – the pressure on pike and perch (fishing, predation, habitat degradation) would have to be reversed to see if the system switched back to the predator-rich state (a reversible phase transition and not alternative states) or did not. Even in the latter case, where a prey-rich state at first appears to be a self-sustaining alternative state that persists after the cessation of pressure on the predators, it may simply be that the prey-rich state proves to be a transient phenomenon (not a regime shift), and the system reverts to the original configuration after a time lag (Frank et al., 2011).

Answer 12: We again thank the reviewer for forcing us to sharpen our argumentation. We agree that the predator-prey reversal effect demonstrated by the path analysis provides a plausible mechanism that may ‘lock’ the system in a stickleback state, but that as long as the drivers causing a decline in predators are not reversed, it is impossible to assess whether the shift is persistent or not. We have now clarified this in the Discussion (lines 435-440).

e) Fig. 5 outlines the putative predator-prey reversal mechanism, along with a trophic cascade. The system is presented as consisting of four trophic levels. However, as noted above, the perturbation that triggered the transition to stickleback dominance came from the fifth trophic level (humans/birds/seals). To really understand whether alternative states operate in this system, observations at contrasting levels of top-down control by this fifth trophic level would be needed. I recognize that these data are likely not available, but the authors should be explicit about that absence, and cautionary that demonstration of alternative states (if that is what is meant by a regime shift) would at the least require comparison across varying pressures in trophic control on perch/pike.

Answer 13: We again agree and refer to the answers above. To better stress what’s certain vs. uncertain in our case study, we have improved Figure 5 by adding the drivers of perch/pike decline and stickleback increase into the right-hand conceptual model ‘stickleback dominance’ (see red text in Fig. 5 conceptual model, lines 454-455).

New version of Figure 5

Figure 5 | A spatial regime shift from perch/pike- to stickleback dominance over ecosystem structure, processes and feedback mechanisms. Center panel shows the influence of year and distance to open sea on the likelihood of perch/pike dominance (1/0; color gradient), after accounting for the influence of wave exposure and latitude. Black dashed line shows the 50% likelihood contour; the point at which an individual bay is as likely to be perch/pike- as stickleback-dominated. Left panel: empirically demonstrated four-level trophic cascade (Eriksson et al. 2009, Donadi et al. 2017) in areas dominated by perch and/or pike (corresponding to blue areas in main panel). Right panel: conceptual model of how stickleback dominance (red areas in main panel) is favored by predator decline and environmental changes (red text), and in turn favors ephemeral algae, by feeding on and controlling crustacean mesograzers (Eriksson et al. 2009, 2011). Abundant stickleback also restrict perch and pike recruitment by feeding on their juvenile stages (Bergström et al. 2015, Byström et al. 2015, Nilsson et al. 2019), which may indirectly benefit stickleback survival through a positive feedback. Arrow thickness is proportional to the strength of interactions.

This paper will make a very valuable contribution, but the central claims are poorly defined and not supported by the data. The paper should be rewritten to focus on the demonstration of predator-prey reversal, and should be much more cautionary about interpreting the evidence as demonstrating a large-scale regime shift.

Answer 14: We sincerely thank the reviewer for very helpful comments overall, including this one. Some of the potential consequences of predator-prey reversal is indeed a key theme of this paper. We, however, still believe that the focus on a spatial regime shift is sufficiently important and novel, even though we cannot (yet) demonstrate whether the shift is truly persistent. The overwhelmingly positive notes by reviewers 1-2 (see above), as well as a few recent studies on spatial regime shifts in high-profile journals (Rocha et al. 2018, Roberts et al. 2019) – that also do *not* test whether the shifts are persistent – strengthen our view. In addition, several studies cited in our manuscript have already described the predator-prey reversal between stickleback on the one hand, and perch (Byström et al. 2015) and pike (Nilsson et al. 2019) on the other.

Finally, the data availability statement (“All data is available upon request from the corresponding author”) is really not up to the community standard any more. Data and code for reproducing the results should be posted in a publicly available repository, and Communications Biology should make this a requirement for publication. This is simply the direction that science is going, and it is a very positive development in terms of guaranteeing reproducibility.

Answer 15: We agree and have now added the data used for the analyses and figures in two supplementary Excel files: Supplementary Data 1 and 2. Moreover, the data is (as described in the manuscript, see lines 627-628) already deposited in a national Swedish database for fish survey data: “The data is compiled in the Swedish national database for coastal fish (<http://www.slu.se/kul>).”

deYoung, B., M. Barange, G. Beaugrand, R. Harris, R. I. Perry, M. Scheffer, and F. Werner. 2008. Regime shifts in marine ecosystems: detection, prediction and management. *Trends in Ecology & Evolution* 23:402–409.

Dudgeon, S. R., R. B. Aronson, J. F. Bruno, and W. F. Precht. 2010. Phase shifts and stable states on coral reefs. *Marine Ecology Progress Series* 413:201–216.

Frank, K. T., B. Petrie, J. A. D. Fisher, and W. C. Leggett. 2011. Transient dynamics of an altered large

marine ecosystem. *Nature* 477:86–89.

Gardmark, A., M. Casini, M. Huss, A. van Leeuwen, J. Hjelm, L. Persson, and A. M. de Roos. 2015. Regime shifts in exploited marine food webs: detecting mechanisms underlying alternative stable states using size-structured community dynamics theory. *Philosophical Transactions of the Royal Society B-Biological Sciences* 370.

Di Lorenzo, E., and M. D. Ohman. 2013. A double-integration hypothesis to explain ocean ecosystem response to climate forcing. *Proceedings of the National Academy of Sciences* 110:2496–2499.

Rocha, J. C., G. Peterson, O. Bodin, and S. Levin. 2018. Cascading regime shifts within and across scales. *SCIENCE* 362:1379+.

Scheffer, M., and S. Carpenter. 2003. Catastrophic regime shifts in ecosystems: linking theory to observations. *Trends in Ecology & Evolution* 18:648–656.

Answer 16: Additional changes made to the manuscript

Modified title: We have modified the title to “A spatial regime shift from predator to prey dominance in a large coastal ecosystem”. This focuses attention to the spatial regime shift, which is the novel aspect. It also reduces the risk of critique on the lack of proof that predator-prey reversal is persistent – i.e. ‘locks’ the ecosystem in an alternative regime – as the points raised by reviewer #3 (see above).

Improved Figure 1a: We have improved Fig. 1a; the conceptual model outlining our main hypothesis of a spatial regime shift. While the old version stressed the hypothesized spatial bistability, the new version emphasizes the spatial regime shift – i.e. that stickleback-dominated areas are expected to have **spread** into the archipelago, at the expense of perch- and pike-dominated areas.

New version of Figure 1:

Figure 1 | Spatial regime shift and bimodality in the coastal Baltic Sea ecosystems. a) Hypothesized spatial transition from a stickleback- to perch/pike-dominated ecosystem regime with increasing distance to open sea at three hypothetical points in time (T_{1-3}), where dotted vertical red lines are spatial breakpoints, and shaded areas in-between are bistable zones. b) Histogram of relative predator dominance (pooling across all bay-years, $N = 829$); an index where 1 = 100% perch/pike domination, and 0 = 100% stickleback domination. P is the likelihood of unimodality based on Hartigan’s dip test, and amplitude (range 0-1) the distinctness of the modes.

Minor textual changes:

Finally, we have made minor textual adjustments here and there to improve flow (see Track changes).

References used in our answers above

- Andersen T, Carstensen J, Hernández-García E, Duarte CM (2009) Ecological thresholds and regime shifts: approaches to identification. *Trends Ecol Evol* 24:49–57
- Bel G, Hagberg A, Meron E (2012) Gradual regime shifts in spatially extended ecosystems. *Theor Ecol* 5:591–604
- Bergström U, Olsson J, Casini M, Eriksson BK, Fredriksson R, Wennhage H, Appelberg M (2015) Stickleback increase in the Baltic Sea - A thorny issue for coastal predatory fish. *Estuar Coast Shelf Sci* 163:134–142
- Byström P, Bergström U, Hjälten A, Ståhl S, Jonsson D, Olsson J (2015) Declining coastal piscivore populations in the Baltic Sea: Where and when do sticklebacks matter? *Ambio* 44:462–471
- Collie JS, Richardson K, Steele JH (2004) Regime shifts: Can ecological theory illuminate the mechanisms? *Prog Oceanogr* 60:281–302
- Conversi A, Dakos V, Gardmark A, Ling S, Folke C, Mumby PJ, Greene C, Edwards M, Blenckner T, Casini M, Pershing A, Mollmann C (2014) A holistic view of marine regime shifts. *Philos Trans R Soc B Biol Sci* 370:20130279–20130279
- deYoung B, Barange M, Beaugrand G, Harris R, Perry RI, Scheffer M, Werner F (2008) Regime shifts in marine ecosystems: detection, prediction and management. *Trends Ecol Evol* 23:402–409
- Donadi S, Austin ÅN, Bergström U, Eriksson BK, Hansen JP, Jacobson P, Sundblad G, Regteren M Van, Eklöf JS (2017) A cross-scale trophic cascade from large predatory fish to algae in coastal ecosystems. *Proc R Soc B Biol Sci* 284
- Eriksson BK, Ljunggren L, Sandström A, Johansson G, Mattila J, Rubach A, Råberg S, Snickars M (2009) Declines in predatory fish promote bloom-forming macroalgae. *Ecol Appl* 19:1975–88
- Eriksson BK, Sieben K, Eklöf J, Ljunggren L, Olsson J, Casini M, Bergström U (2011) Effects of Altered Offshore Food Webs on Coastal Ecosystems Emphasize the Need for Cross-Ecosystem Management. *Ambio* 40:786–797
- Nes EH van, Scheffer M (2005) IMPLICATIONS OF SPATIAL HETEROGENEITY FOR CATASTROPHIC REGIME SHIFTS IN ECOSYSTEMS. *Ecology* 86:1797–1807
- Nilsson J, Flink H, Tibblin P (2019) Predator-prey role reversal may impair the recovery of declining pike populations. *J Anim Ecol*:1365-2656.12981
- Roberts CP, Allen CR, Angeler DG, Twidwell D (2019) Shifting avian spatial regimes in a changing climate. *Nat Clim Chang* 9:562–566
- Rocha JC, Peterson G, Bodin Ö, Levin S (2018) Cascading regime shifts within and across scales. *Science (80-)* 362:1379–1383
- Scheffer M, Carpenter SR (2003) Catastrophic regime shifts in ecosystems: linking theory to observation. *Trends Ecol Evol* 18:648–656

REVIEWERS' COMMENTS:

Reviewer #1 (Remarks to the Author):

I have read the rebuttal letter from the authors and the revised manuscript. I think the authors have addressed the issues raised by the reviewers satisfactory. I will therefore recommend the manuscript to be published in Communications Biology.

Reviewer #3 (Remarks to the Author):

I thank the authors for their carefully considered responses to my earlier criticisms of the paper. I recommend publication of the manuscript in its current form.